# R4Alz-Revised: A Tool Able to Strongly Discriminate ‘Subjective Cognitive Decline’ from Healthy Cognition and ‘Minor Neurocognitive Disorder’

**DOI:** 10.3390/diagnostics13030338

**Published:** 2023-01-17

**Authors:** Eleni Poptsi, Despina Moraitou, Emmanouil Tsardoulias, Andreas L. Symeonidis, Vasileios Papaliagkas, Magdalini Tsolaki

**Affiliations:** 1School of Psychology, Faculty of Philosophy, Aristotle University of Thessaloniki (AUTh), 54124 Thessaloniki, Greece; 2Laboratory of Neurodegenerative Diseases, Center for Interdisciplinary Research and Innovation, Aristotle University of Thessaloniki (CIRI—AUTh), 54124 Thessaloniki, Greece; 3Day Center “Greek Association of Alzheimer’s Disease and Related Disorders (GAADRD)”, 54643 Thessaloniki, Greece; 4School of Electrical and Computer Engineering, Faculty of Engineering, Aristotle University of Thessaloniki (AUTh), 54124 Thessaloniki, Greece; 5Department of Biomedical Sciences, International Hellenic University, 57001 Thessaloniki, Greece; 61st Department of Neurology, Medical School, Aristotle University of Thessaloniki (AUTh), 54124 Thessaloniki, Greece

**Keywords:** early diagnosis, neurodegeneration, Subjective Cognitive Decline, R4Alz-R battery

## Abstract

Background: The diagnosis of the minor neurocognitive diseases in the clinical course of dementia before the clinical symptoms’ appearance is the holy grail of neuropsychological research. The R4Alz battery is a novel and valid tool that was designed to assess cognitive control in people with minor cognitive disorders. The aim of the current study is the R4Alz battery’s extension (namely R4Alz-R), enhanced by the design and administration of extra episodic memory tasks, as well as extra cognitive control tasks, towards improving the overall R4Alz discriminant validity. Methods: The study comprised 80 people: (a) 20 Healthy adults (HC), (b) 29 people with Subjective Cognitive Decline (SCD), and (c) 31 people with Mild Cognitive Impairment (MCI). The groups differed in age and educational level. Results: Updating, inhibition, attention switching, and cognitive flexibility tasks discriminated SCD from HC (*p* ≤ 0.003). Updating, switching, cognitive flexibility, and episodic memory tasks discriminated SCD from MCI (*p* ≤ 0.001). All the R4Alz-R’s tasks discriminated HC from MCI (*p* ≤ 0.001). The R4Alz-R was free of age and educational level effects. The battery discriminated perfectly SCD from HC and HC from MCI (100% sensitivity—95% specificity and 100% sensitivity—90% specificity, respectively), whilst it discriminated excellently SCD from MCI (90.3% sensitivity—82.8% specificity). Conclusion: SCD seems to be stage a of neurodegeneration since it can be objectively evaluated via the R4Alz-R battery, which seems to be a useful tool for early diagnosis.

## 1. Introduction

The diagnosis of the minor neurocognitive diseases in the clinical course of dementia before the clinical symptoms’ appearance is nowadays the holy grail of neuropsychological research. Thus, Subjective Cognitive Decline (SCD) seems to be a stage in neurodegeneration prior to Mild Cognitive Impairment (MCI) [1,2,3] which lasts about 15 years [1], whilst the risk of people with SCD to convert either to MCI or to major neurocognitive disorders such as Alzheimer’s Disease ranges from 4.5–6.5 times higher than the risk of normal older adults [1,4]. Despite the fact that until today no universally sensitive and accepted cutoff scores exist, to discriminate healthy cognition from SCD, and SCD from MCI, in the last years, significant progress exists. Since there are studies showing that the brief cognitive tasks used in primary care cannot differentiate SCD from MCI with high accuracy [5,6,7], comprehensive neuropsychological test batteries with age-, gender-, and education-adjusted normative data are proposed [8].

Nowadays, there are studies attempting to classify the SCD by using standardized validated tests. Such a battery is the “REMEDES for Alzheimer” battery (R4Alz), which attempts to differentiate adults with SCD from cognitively healthy adults (young, middle-aged, and older), and people with MCI as well [9]. The battery was initially designed to assess cognitive control abilities, via subtasks that assess working memory components, attention control abilities, inhibition and switching, and cognitive flexibility [10]. The first pilot study of the R4Alz was performed in 2020 in 175 Greek adults, and had as its main target to examine if the battery could successfully detect SCD, and differentiate it from the spectrum of cognitively healthy status and MCI. According to the results, and despite the fact that, overall, the seven tasks showed a significant differential ability among groups, however, only three out of them, as well as one subtask, showed a significant differential potential between SCD and healthy cognition in older age. These were: (a) the Working Memory (WM) Updating and the WM component of the Episodic Buffer, (b) the Inhibition plus the Task/Rule Switching task, (c) the failed sets on the inhibition subtask and the task/rule switching, and (d) the Cognitive Flexibility task. The Receiver Operating Characteristic (ROC) curve analysis, used for assessing the predictive value of the aforementioned, showed that there was an excellent discrimination (AUC 0.991) of SCD from healthy cognition in young adults with a high sensitivity (100%) and specificity (97.6%), and in middle-aged adults (AUC 0.981) with also a high sensitivity (100%) and specificity (93.5%). Furthermore, the four tasks could also differentiate (AUC 0.960) SCD from healthy cognition in older adults with a sensitivity of 85.3% and a specificity of 92.9%. Unfortunately, the discrimination of SCD from MCI was just fair (AUC 0.758) with a sensitivity of 76.6% and a specificity of 73.5% [9].

The importance of the above findings is that, to the best of our knowledge, R4Alz is the first battery which provides cutoff scores with such high rates of sensitivity and specificity, discriminating with an excellent rate cognitively healthy adults from people with SCD. Therefore, via the above study, it is clear that SCD is not subjective but is a new stage of neurodegeneration, providing the clinicians with more accurate criteria to define SCD, that comprise not only the criteria proposed by the SCD working group [3], but also objective, quantitative specific indicators and cutoff scores.

Despite the fact that the R4Alz battery seems to have an excellent discrimination ability between healthy cognition and SCD, it has only a fair discrimination ability between SCD and MCI. The possible causes for the lack of sensitivity are potentially correlated with the nature of the tasks that have been selected at baseline, which were developed to examine cognitive control abilities on the basis of the extant literature on SCD [11,12]. People with SCD seem to complain about cognitive control deficits such as shifting of attention, inhibition, and in general, deficits involving executive function abilities. A few examples that people with SCD refer to include a situation in which they are engaged in an activity and forget what their previous activity was (what was I doing before answering the phone call?) or a situation in which they have difficulties concentrating and reading a book because they hear the neighbor’s doorbell ringing. Moreover, according to publications, there are people with an SCD score lower than healthy controls in cognitive control abilities [13], despite the fact that their scores are in healthy ranges. Therefore, cognitive control abilities were chosen to be included in the battery. However, even though the original R4Alz tasks evaluated cognitive control abilities as a part of the executive system spectrum (which seems to be deficient early in the course of neurodegeneration), people with MCI seem to have more deficits in memory capacity and especially in episodic memory (EM) [14]. Unfortunately, this ability was excluded from the original battery design, which could be one of the reasons for the low differentiation rates of people with SCD from MCI.

As far as episodic memory (EM) is concerned, it is an ability closely related to executive functions [15,16,17]. During the last years, there are studies that attempted to examine the relation between executive functions and EM, as well as the specific mechanisms underlying the interaction. Episodic memory is one of the most important neurocognitive memory systems [18], defined as the ability to remember events that were taking place in a certain situation at a certain time [19]. Episodic memory seems to be the first memory system that declines both in normal and in pathological aging such as in Alzheimer’s disease and it is quite sensitive to the effects of age [20]. According to the executive decline hypothesis [21], the memory decline which appears in older adults in comparison to younger adults is possibly associated with a decline in executive function abilities [15,21]. The study of Clarys et al., in 2009 [16], utilized the theoretical model of Miyake (2000) [22] regarding cognitive control, to investigate whether deficits in updating, shifting, and inhibition of a prepotent response are associated with episodic memory deficits. The participants of the study were 88 people, categorized into two groups (young vs. older adults). Participants had to perform (a) a recognition memory test by using the Remember/Know/Guess method and (b) executive function tests which assess cognitive control abilities. During the recognition memory test, a list of taxonomically unrelated concrete words was presented. During the encoding phase participants had to read the words aloud and to remember them for a later test. Afterwards, during the recognition task, the participants were asked to recognize the previously presented words from another list containing the initial words and distractors. For each recognized word, they had to indicate if their response was based on Remembering (R), Knowing (K), or Guessing (G). According to their results, specific executive deficits account for the age-related decline in memory performance. Specifically, in their study there was a strong correlation between all cognitive control abilities [22] but the strongest correlation was noted between the updating with the Remember response measures and the episodic memory. The above correlation was also found to be larger for the older adult group than the younger one. As the authors support, the updating process is involved in the elaboration and control of mental representation in memory during the presentation of the word list and during the recognition stage. Therefore, via this procedure, suitable strategies are being used during the encoding and recognition phase for improving the memory trace, and this mechanism is possibly not only to improve the learning and retrieval of information, but also the associated contextual details needed for recognition, based on conscious recollection [16].

Similarly, the study of Yuan et al., in 2016 [17], was performed in an amnestic MCI group and healthy controls, utilizing resting-state functional magnetic resonance imaging (fMRI) scans for assessing the potential association between the executive and episodic memory networks. According to their results, there was an EM and executive function correlation both for the groups of healthy and MCI, and specifically, it was found that the executive function network may mediate episodic memory performance in people with MCI [17], proving the notion that executive function is closely associated with episodic memory loss.

On the other hand, as far as the SCD is concerned, it seems that there are studies indicating difficulties in EM. The subjective cognitive complaints usually include memory and language abilities, apart from executive functions [11].

Consequently, since cognitive control and EM are both impaired in amnestic MCI [23], and are correlated with an increased rate of progression in dementia [24], we considered that the R4Alz battery should be expanded by using an extra EM task. Moreover, we considered that a different and more demanding task of switching and especially cognitive flexibility would also help the R4Alz to differentiate with more accuracy people with SCD from people with MCI, and from cognitively healthy older adults.

## 2. The Purpose and the Hypotheses of the Study

The purpose of the present study was to extend the R4Alz battery by designing and performing an extra EM task as well as an extra cognitive control task, comprising a combination of a shifting of attention and cognitive flexibility subtask, for enhancing the ability of the R4Alz to discriminate SCD from healthy cognition in advanced age (HC), and from MCI.

To achieve the main purpose, the study aimed to examine whether the performance on the R4Alz’s tasks (the old plus the new tasks) is affected by age and education, as well as to examine whether the new R4Alz battery as an entity (basic tasks plus the new) can clearly discriminate SCD from HC in advanced age and MCI.

Therefore, the study’s hypotheses were formulated as follows:Hypothesis 1: The battery’s subtasks would not be affected by demographic variables such as age or education.Hypothesis 2: The extended R4Alz battery, namely, the R4Alz-Revised (R4Alz-R), would adequately differentiate adults of advanced age with SCD from healthy controls and people with MCI, as well as healthy controls from people with MCI.

## 3. Method

### 3.1. Design

The study’s design comprised three diagnostic groups: (a) community-dwelling cognitively healthy adults of advanced age (HC, ≥50 years), (b) people with SCD [3], and (c) people diagnosed with MCI [25].

In the present study, people with SCD were our main target group, since we are attempting to define early indicators of cognitive decline as soon as possible. Therefore, the results for the SCD group with the other two (HC-MCI) will be analyzed. An extra analysis will compare HC and MCI groups, to investigate whether the battery’s tasks differentiate them. Furthermore, a new methodology of creating mathematical formulae for the battery’s total score will be used, in order to explore which is the best way to combine individual tasks’ scores, so as to discriminate SCD from HC and MCI, and HC from MCI. Each of these formulae will comprise only the tasks that have statistical significance dependent on diagnosis and not on age or educational level.

### 3.2. Ethics

All the participants were orally and in written form informed of the purpose of the study and had the opportunity to ask questions. They were also informed that their data would be confidentially collected in an electronic database. The participants gave written consent at the time of their visit, agreeing that their participation was voluntary and that they could withdraw at any time, without giving a reason and without cost. Due to the specific type of the current research, demographic data such age, gender, or occupation were selected. Since these are considered personal data, the European Union law that exists since 28 May 2018 was applied. According to the law, the use of sensitive personal data is allowed only for research reasons. Therefore, the participants were informed accordingly and they also agreed that their personal data could be deleted from the web-database after a written request. The study’s protocol was approved by the Scientific and Ethics Committee of the Greek Association of Alzheimer’s Disease and Related Disorders, and followed the principles outlined in the Helsinki Declaration.

### 3.3. Participants

The study sample consisted of volunteers, (a) cognitively healthy adults (HC), as well as (b) people with SCD, and (c) persons with MCI, recruited from the broad area of Thessaloniki. The sub-sample of cognitively healthy adults was recruited via (a) an invitation that was posted on social media (Facebook pages) and, (b) an associated publication in press. The sub-samples of people with SCD and MCI were visitors to the Day Care Centre “Saint Helen” of the Alzheimer Hellas (DCCAH), in Thessaloniki, during the period of October 2021 to April 2022, who visited the DCCAH for a yearly medical and psychological check-up routine.

The study sample included 80 people. The participants were categorized in: (a) HC (n = 20, 6 men and 14 women, age range: 52 to 73 years, M = 61.60, SD = 6.58, education range: 12 to 24 years, M = 16.30, SD = 3.04); (b) SCD, (n = 29, 9 men and 20 women, age range: 50 to 86 years, M = 61.17, SD = 7.00, education range: 6 to 24 years, M = 13.31, SD = 4.20); and (c) people with MCI, (n = 31, 7 men and 24 women, age range: 51 to 82 years, M = 68.67, SD = 8.43, education range: 6 to 23 years, M = 13.45, SD = 4.31).

The Pearson’s chi square test was used for examining the gender differences between groups. The analysis showed that the three groups did not differ in gender, χ 2 (2, 80) = 733, *p* = 0.771. As far as age and education is concerned, ANOVAs showed that, the groups differed in age, F2, 80 = 3.656, *p* = 0.030, and education, F2, 80 = 5.230, *p* = 0.007. Regarding age, the Scheffé post hoc comparisons showed that HC significantly differed from MCI, I-J = −0.43, *p* < 0.05, but not from SCD, I-J = −0.39, *p* > 0.05. HC group was younger than people with MCI. As far as education is concerned, the Scheffé post hoc comparisons showed that SCD significantly differed from HC, I-J = −0.65, *p* < 0.05, but not from MCI, I-J = −0.00, *p* > 0.05. HC group also significantly differed from MCI group, I-J = 0.64, *p* < 0.05, with HC having more years of education compared to SCD and MCI. Study sample characteristics are presented in Table 1.

### 3.4. Exclusion Criteria

The exclusion criteria for all groups were: (a) history of psychiatric illness or affective disorder (Major Depression/General Anxiety Disorder); (b) substance abuse or alcoholism; (c) history of traumatic brain injury; (d) brain tumor, encephalitis, epilepsy history, Parkinson’s disease, stroke history, and other neurological disorders such as hydrocephalus; (e) cancer in the last 5 years, myocardial infarction in the last 6 months, or pacemaker; (f) thyroid issues or diabetes; (g) drug treatment with opioids, B12, folate, or thyroid; (h) sensory deficits; (i) drug treatment with opioids or medication for B12 vitamin deficiency; and (j) absence of subjective cognitive complaints (except from the people with SCD).

The reason for this extended exclusion criteria were to ensure to the largest possible extent that the cognitively healthy control group as well the SCD group does not have factors that can cause cognitive problems. Moreover, all participants followed an extended neuropsychological assessment: For the exclusion of affective disorders, the Geriatric Depression Scale and the Beck Depression Inventory [26,27,28], and the Short Anxiety Screening Test [29,30] and the Beck Anxiety Inventory [31] were used. The Neuropsychiatric Inventory [32,33] was also used for the exclusion of neuropsychiatric symptoms. The Mini Mental State Examination [34,35], as well as the Montreal Cognitive Assessment Scale [36,37], were also used. The Functional Cognitive Assessment [38] was used in order to assess their ability to organize and execute 6 different Activities of Daily Living. Furthermore, standardized tests for the assessment of general cognitive and functional abilities, memory capacity, language abilities, executive functions, and attention were used as well. The Global Deterioration Scale (GDS) [39] was used to determine a patient’s status, with regard to the progression of their disease. Therefore, according to GDS, at stage 1 were categorized people with no cognitive decline and normal functioning, with no deficits at all; people with subjective cognitive complaints who express worries regarding their symptoms are categorized in stage 2; whereas people with MCI are categorized in stage 3. Finally, the first three questions of the Subjective Cognitive Decline Questionnaire (SCD-Q) [40] were also used in order to evaluate any complaints regarding cognitive function in HC and people with SCD. The entirety of the neuropsychological tests included in the battery is presented in detail in Tsolaki et al. in 2017 [41].

### 3.5. Inclusion Criteria

The inclusion criteria regarding HC adults comprised men and women aged above 50 years, with 6 and above years of education, without subjective cognitive complaints (negatively answer to the first 3 questions of the Subjective Cognitive Decline Questionnaire (SCD-Q)) [40], and stage 1 of the disease according to Global Deterioration Scale [39].

Regarding people with SCD, the inclusion criteria were based on the diagnostic criteria proposed by SCD-I Working Group [3] and comprised: (a) feelings of worse memory performance, not associated with the presence of depressive symptoms; (b) absence of objective cognitive deficits, according to the neuropsychological tests; and (c) stage 2 of the disease according to Global Deterioration Scale [39]. Moreover, all participants had to positively answer to the first 3 questions of the Subjective Cognitive Decline Questionnaire (SCD-Q) [40] to adhere to the SCD criteria published by Jessen et al. in 2014 [3]: (1) Do you perceive memory or cognitive difficulties? (2) Would you ask a doctor about these difficulties? (3) In the last two years, has your cognition or memory declined?

Finally, as far as people with MCI are concerned, the inclusion criteria were based on the DSM-5 criteria for Mild Neurocognitive Disorders [25]. Their diagnosis was supported by neurological examination, neuropsychological and neuropsychiatric assessment, neuroimaging (computed tomography or magnetic resonance imaging), and blood tests, by a consensus of specialized health professionals of Alzheimer Hellas, considered experts in neurocognitive disorders. The inclusion criteria comprised: (a) diagnosis of Minor Neurocognitive Disorders according to DSM-5, (b) Mini-Mental State Examination (MMSE) total score ≥ 24, (c) stage 3 of the disease according to Global Deterioration Scale, and (d) 1.5 standard deviation (SD) below the normal mean according to age and education, in at least one cognitive domain according to the utilized neuropsychological tests.

### 3.6. Tools

#### 3.6.1. The R4Alz Battery via the Physical, Three-Dimensional Devices (REMEDES Pads)

All the tasks of the baseline battery were chosen to be performed during the present study, for examining whether the battery (comprising the baseline tasks plus the new) can discriminate the three groups of adults. The baseline tasks were as follow: (a) Working Memory Capacity and Updating Task (WMCUT): Subtask 1 (working memory component of short-term store) (WMCUT Subtask 1), (b) WMCUT Subtask 2 (working memory component of central executive), (c) WMCUT Subtask 3 (working memory updating & working memory component of the episodic buffer); (d) Attentional Control Task (ACT), (e) Inhibitory Control & Task/Rule Switching Task (ICT/RST): Subtask 1 (Inhibition) and Subtask 2 (Task/Rule switching) ICT/RST 1 & 2; For this task were additionally measured the total number of errors of Inhibition and Task/Rule switching (ICT/RST 1 & 2 SE), the Failed sets of the Inhibition and Task/Rule switching subtask (ICT/RST 1&2 FS); (f) Cognitive Flexibility Task (CFT) (see for details Poptsi et al., 2019) [10].

#### 3.6.2. The Procedure of Developing the New Digital-Designed and -Performed R4Alz’s-R Tasks

As far as the new tasks that were integrated into the already existing R4Alz battery are concerned, they were designed for the needs of the best discrimination between SCD and healthy controls, and between SCD and MCI. However, the original form of R4Alz did not allow one to utilize a variety of pictures/photos in the system that was the prerequisite for the development of a new memory task, and this would require different hardware development of the REMEDES pads (e.g., adding a touch screen). Therefore, the digital design and evaluation was adapted. However, the constructors attempted to adapt the rationale of the initially designed format of the R4Alz battery (pads) whenever possible (Figure 1).

It should be stated that in consistence with the original battery, the digitally extended tasks were also based on well-known and commonly utilized tests. To be more precise, the extra tasks were developed based on the following tests: (a) Doors and People [42] for accessing EM, and (b) Design Fluency of the Dellis and Kaplan battery (D-KEFS) [43] for cognitive flexibility.

After the initial digital design of the extra R4Alz tasks, a procedure of a preliminary evaluation was performed with a limited number of occasionally random participants (5 healthy middle-aged adults, 5 older adults, 5 people with SCD, and 5 people with MCI) to test (a) whether the new tasks were understandable and clear for the examinees, and b) whether the tasks were too simple or too difficult to be performed by examinees. During the initial observation, several technical and methodological errors were noticed, and therefore the needed corrections were performed for greater reliability and validity.

Afterwards, during this study, the new digital tasks were administered. The administration of the new tasks lasted for almost half an hour, whereas the baseline task’s duration was almost an hour. The order of tasks’ administration differed from participant to participant to avoid potential order effects on performance. However, at this point, it should be mentioned that the R4Alz battery via the REMEDES pads was administered first, followed by its new digital revision part.

### 3.7. Description of the New Tasks

By using the original design of the battery, before each task and after its written description, a short example is provided, to make sure that the participant understands the task. The new tasks are the following:

#### 3.7.1. Cognitive Flexibility Task Part 2 (CFT2): Inhibitory Control plus Task/Rule Switching

This task is the second part of Cognitive Flexibility. The first part exists in the original R4Alz battery. The Cognitive Flexibility Task Part 2 comprises 4 different subtasks with conditions with raising degree of difficulty. These tasks assess either Inhibitory Control or Task/Rule Switching, or both (cognitive flexibility). The exact abilities required for each condition’s implementation are as follows.

Condition A (1st level of difficulty):

In this subtask, seven green and red REMEDES pads are presented. In Condition a, the participant is asked to perform the following steps in an iterative order:Step 1: All the **red** pads starting from the **left** and moving to the **right**Step 2: All the **green** starting from the **right** and moving to the **left**Step 3: All the **red** pads starting from the **right** and moving to the **left**Step 4: All the **green** pads starting from the **left** and moving to the **right**

The participants are asked to follow the baseline instructions by alternating the steps until the Condition is over. Therefore, condition a requires the examinee to monitor and deactivate a pad of a certain color which is different for each step. During the monitoring, a second color is also presented. As a consequence, the participant has to inhibit the second color that is presenting by avoiding to deactivate it. Furthermore, the task requires switching of attention, since when each set of seven pads is over, the examinee is asked to switch his attention and to deactivate the other color, while simultaneously switching the direction of the deactivations (e.g., from left to right or from right to left).

The rating of this condition is the number of erroneous sets. Since the condition comprises eight sets, the maximum number of errors is eight (8). On the other hand, the minimum number of errors is equal to zero (0), if all the answers of the examinee to the condition a are correct (Figure 2).

Condition B (2nd level of difficulty)

The participant is asked to deactivate the following:Step 1: Deactivate green and red pads from left to right, with alternating colors, skipping continuous occurrences of the same color, starting from green (green, the next red, the next green, etc.)Step 2: Deactivate green and red pads from right to left, with alternating colors, skipping continuous occurrences of the same color, starting from red (red, the next green, the next red, etc.)

The participants are asked to follow the baseline instructions by alternating the steps until the condition is over. The examinees in condition B are asked to deactivate not only the red pads during a set, but both green and red pads in alternative order, beginning from the left to right. Therefore, the participants have to inhibit the previous automatic responses required in condition A, and furthermore, to skip and inhibit the consecutive occurrences of the same color. Condition B also requires switching of attention, since the examinee is asked to alter the baseline instructions (after the first set) by means of starting color and direction. Similarly to condition A, condition B also comprises eight sets and therefore, the rating scores are between eight (8) and zero (0), where eight is the maximum number of errors, when the examinee makes an error in each set, and zero when all sets are performed correctly.

Condition C (3rd level of difficulty):

The participant is asked to follow the below 4 steps:Step 1: Deactivate green and red pads from left to right, with alternating colors, skipping continuous occurrences of the same color, starting from red (red, the next green, the next red, etc.)Step 2: Deactivate green and red pads from right to left, with alternating colors, skipping continuous occurrences of the same color, starting from green (green, the next red, the next green, etc.)Step 3: Deactivate green and red pads from right to left, with alternating colors, skipping continuous occurrences of the same color, starting from red (red, the next green, the next red, etc.)Step 4: Deactivate green and red pads from left to right, with alternating colors, skipping continuous occurrences of the same color, starting from green (green, the next red, the next green, etc.)

Condition C is similar to condition B; however, in this condition, the participants are asked not only to alter and shift their attention from color to color, but also to shift the direction of deactivations from the one set to another. Therefore, the participants are asked to follow 4 different steps/patterns of altering colors and direction of deactivation (from right to left and from left to right). Moreover, inhibition is also required. Participants are asked to skip continuous occurrences of the same color among every set. The condition is over when the examinees complete all conditions/steps until the condition is over. In this condition, the maximum number of errors is also eight (8), equal to the number of the 8 sets, whilst the minimum number of errors is also equal to zero (0).

Condition D (Visual fluency task—Combinations—VFT: Planning, Inhibitory Control, and Visual Fluency):

The participant is asked to continuously deactivate all four green pads with a different order each time, i.e., to find as many different combinations as possible. For example, in the case where only 2 pads were presented, the participant could turn off the pads following the order 1-2 and then following the order 2-1. The participant is asked to attempt to turn off the 4 presented pads by implementing as many different combinations as they can in 60 s. Here, abilities of initiation of problem solving are required. The task also requires from the examinee fluency in generating visual patterns and inhibition of previously executed responses. In this condition the maximum number of the correct answers are 24, which corresponds to 24 combinations, whereas the minimum number is zero (0).

#### 3.7.2. Episodic Memory Task—Windows (EMT-W)

Fifteen windows are presented to the examinees in a row. The examinee is asked to observe each window very carefully and with great accuracy by emphasizing the windows details. The examinee is informed that there is no time limit and therefore they can observe for as much time as needed until they are sure that they have kept the images of each window in their mind. After the presentation of 15 windows, 15 more screens appear, including four similar windows; for each screen, one is the original window (one of the 15 presented) and 3 are distractors (new windows), which differ between each other in details. The examinee is asked to recognize the window that they have initially seen. The maximum number of errors is fifteen (15), equal to the number of the windows presented initially, whilst the minimum number of errors is equal to zero (0) (Figure 3).

Condition B (2nd Level of Difficulty)

In this condition, the same procedure is followed; nevertheless, the differences are smaller, thus harder to detect (Image 5). Again, the maximum number of errors is fifteen (15), equal to the number of the windows, whilst the minimum number of errors is equal to zero (0).

Hence, with regard to the total battery, the R4Alz-R comprises 14 separate sub-scores that are categorized based on the abilities that the battery assesses. These scores include the total scores of (a) the WMCUT Subtask 1 and 2 and 3; (b) the total score of ACT; (c) the total scores of ICT/RST Subtask 1 and 2, as well as the total number of errors of ICT/RST 1 and 2, and the Failed sets (as two separate sub-scores); (d) the CFT; (e) the sub-scores of conditions a, b, c, and d (VFT) of the CFT2; and (f) the 2 sub-scores of the 2 conditions of the windows tasks. The total score of the battery that is potentially capable of discriminating between groups will be defined after the separate analysis of each task and will be described in the Section 4.

The electronic form of the battery exists in the following link, where both the original R4Alz and the revised R4Alz-R tests can be deployed/administered: http://r4alz-online.issel.ee.auth.gr/, accessed on 15 July 2022.

### 3.8. Statistical Analysis

For data analysis, the IBM SPSS (version 23.0; IBM Corp, Armonk, NY, USA) [44] and the JASP (version 16) [45] were used.

Mediation analysis deployed in JASP declares how a prognostic variable is related to an outcome variable, indicating that the relationship between two variables is affected by a third variable called mediator [46,47]. Direct and indirect effects emerge from mediation analysis. As a direct effect, the relation between the predictor variable and the outcome variable is defined, and as indirect effect, we consider the effect of the predictor on the outcome through the mediator [46,47]. Mediation analysis was used to examine whether and to what extent diagnostic group (the predictor) affects directly or/and indirectly—via age and education—the performance on the R4Alz-R subtests. Bootstrapping was used to examine the significance of the indirect effect. Indirect effects were computed for each of 1000 bootstrapped samples [45,48]. Mediation analysis in JASP is performed in a structural equation context: mediation models are path models including mediators.

With regard to the formulation of the mediation models, we were based on the “vascular hypothesis of cognitive aging” theory [49]. According to this, age is a descriptive index associated with cognitive decline due to the fact that vascular pathology and neurodegenerative conditions increase as one gets older. Vascular or/and neurodegenerative pathologies set limits on the compensatory function of cognitive reserve in cognitive decline. Therefore, the role of education as an important dimension of cognitive reserve could be limited by these pathologies. Based on the aforementioned, we considered “diagnosis” as the predictor variable in our mediation models, since it refers to the underlying pathology. In the same context, Age and Education were considered as potential mediators, since they are factors that could be affected by the underlying pathology and at the same time, they are associated with cognitive performance. R4Alz-R scores were set as outcome variables.

An alpha level equal to 0.004 (0.05/13) was adopted to avoid problems related to multiple testing.

The receiver operating characteristic curve (ROC curve) analysis was also used for assessing the predictive value of the R4Alz-R subtasks to discriminate SCD from MCI and from HC, as well as MCI from HC. The cut-off points were determined by maximizing the Youden index [50]. The area under the curve (AUC) of the ROC curve was used to quantify the R4Alz-R discriminant potential in fair, good, perfect, or excellent according to the relative literature (AUC values from 1.0 are perfect, 0.9–0.99 is excellent, 0.8–0.89 is good, and 0.7–0.79 is fair, and 0.51–0.69 is a poor test) [51,52].

At this point, it is worth mentioning that, to investigate the best score to discriminate the three groups, a specific procedure was used. This procedure depends on the fact that there are many mathematical ways to combine a battery’s scores or subtasks which seems to have better discriminant ability by several ways for maximizing the discriminant validity of a battery. The specific procedure that was used for the needs of the present study is presented in the Section 4.

## 4. Results

### 4.1. Mediation Analyses

As already mentioned, in mediation analyses, age and education were defined as mediators, subtests’ performance as the outcome variables, with diagnostic group (HC, SCD, MCI) as the predictor. Mediation analyses were performed separately for (a) SCD and HC groups, (b) SCD and MCI groups, and (c) HC and MCI groups.

(a)Mediation analysis in SCD and HC groups

As far as the comparison between the SCD and HC groups is concerned, the mediation analysis showed that there was a significant direct effect of diagnosis on performance in the WMCUT Subtask 3, ICT-RST 1&2, ICT-RST FS, CFT, and CFT condition b (Table 2a). Diagnosis was not found to have any significant indirect effects—via age or education—on those tasks’ performances (Table 2b).

(b)Mediation analysis in SCD and MCI groups

Regarding the comparison between the SCD and MCI groups, the mediation analysis showed that the direct effect of diagnosis on performance in ICT-RST 1&2, ICT-RST SE, in CFT2 Condition a, as well as in EMT-W condition a, was significant (Table 3a). On the other hand, diagnosis was not found to display any significant indirect effect—via age and education—on the aforementioned subtasks’ and tasks’ performances (Table 3b).

(c)Mediation analysis in HC and MCI groups

Regarding the comparison between HC and MCI, it seems that according to the mediation analysis, the direct effect of diagnosis on WMCUT Subtask 2 and WMCUT Subtask 3 performance, on ACT, on ICT-RST 1 & 2, on ICT-RST SE and ICT-RST FS, on ICT-RST 3, on CFT2 Condition a, b, and d (VFT), as well as on EMT-W condition a and EMT-W condition b, was significant (Table 4a). Diagnosis was not found to display any significant indirect effect—via age and education—on all the aforementioned subtasks’ performances (Table 4b).

### 4.2. Discriminant Validity

#### 4.2.1. R4Alz-R Scoring

The aforementioned analysis showed that there are a number of subtasks that show a significant statistical difference in each pair of groups due to diagnosis (health, subjective cognitive decline, and mild cognitive impairment). This means that, for example, there is a different set of subtasks on which only direct effects of the diagnostic group, without any mediation of age or education, was found for the SCD and HC subsamples, in comparison to the SCD and MCI groups. To examine if the R4Als-R battery is able to discriminate between healthy cognition, subjective cognitive decline, and mild cognitive impairment, a single score must be calculated from the individual subtasks’ scores, which will be utilized to calculate a cutoff value. Nevertheless, it is obvious that there is an infinite number of ways to mathematically combine the scores of a set of subtasks into one overall score, each of which will generate different cutoff values, and subsequently different sensitivity and specificity scores. In this section, we attempt to propose a methodology via which a number of overall scores can be generated from individual variables (subtasks scores), so as to investigate which has the best discriminant potential.

Step I—Variables selection

Let us assume that our analyses showed that for $G_{A}$ (group A) vs. $G_{B}$ (group B), there are N subtasks that offer a significant statistical difference, namely, $X_{1},X_{2}\ldots X_{N}$, forming the Xset. Obviously, it makes sense to use only these subtasks for generating a final score, since they can best discriminate $G_{A}$ from $G_{B}$.

Step II—Preprocessing

Since the scores of the $X_{i}$ subtasks will participate in a single mathematical formula which will be the final score, it makes sense to normalize each of the sub-scores in X, so as to have a common range. The easiest approach is to normalize all $X_{i}$ using the min-max normalization technique, as such:$$\bar{X_{i}}=\frac{X_{i}-min\left(X_{i}\right)}{max\left(X_{i}\right)-min\left(X_{i}\right)}$$

This step generates an $X_{n}$ set, comprising all the normalized variables that present a significant statistical difference, now ranging from $\left[0,1\right]$.

Step III—Scoring mathematical formulation

After the preprocessing step, we are ready to create a mathematical formula which will combine all normalized scores, so as to produce a final score. Next, some approaches are described:

Approach A: Sum of scores

Inarguably, the simplest approach to combine a set of variables into a single one is calculating the sum (Σ) of all involved scores, i.e.,
$$S_{\Sigma }=\overset{N}{\underset{i=0}{\sum }}\bar{X_{i}}$$

Since all scores are bounded to $\left[0,\;1\right]$, all variables will participate with the same weight/importance towards generating the final score.

Approach B: Sum of squared scores

Since all scores are normalized to $\left[0,\;1\right]$, we can take advantage of this fact by calculating the sum of squared scores, i.e.,
$$S_{\Sigma^{2}}=\overset{N}{\underset{i=0}{\sum }}{\bar{X_{i}}}^{2}$$

As we know, if we square a value that exists between $\left[0,\;1\right]$, this value still remains in the same range (thus the variable remains normalized), but at the same time this non-linear operation lowers each value by a magnitude that is related to the value’s distance from 1. In simpler words, values that have a large distance from 1, after it is squared, tend to be significantly reduced (e.g., 0.5^2^ = 0.25, a reduction of 50%), whereas values near 1, if squared, tend to be slightly altered (e.g., 0.9^2^ = 0.81, a reduction of 10%). If we think of the actual meaning of our variables, the ones with high values (near 1) will be the ones from subtasks where the cognitively “better” group achieved maximum performance, whereas low values (near 0) are probably individuals from the cognitively “worse” group, whose performance lies in the lowest possible score. Conclusively, by raising all of our variables to the square power, we further widen the performance gap between the cognitively “good” group and cognitively “bad” one.

Approach C: Sum of squared scores, scaled by AUC

At this point, all of our variables are normalized to $\left[0,\;1\right]$ and raised to the square power, maintaining them normalized in the same range. Nevertheless, this may not be an optimal strategy, since some variables better discriminate $G_{A}$ from $G_{B}$; thus, it would make sense to further boost the scores of these variables against all the rest. One way to objectively calculate the discrimination capacity of each variable is by performing a ROC curve analysis and using the AUC value (Area Under the Curve) which is bound to $\left[0,\;1\right]$, and a high value indicates a high discriminant potential. Therefore, this approach’s score is formulated as such:$$S_{\Sigma^{2}AUC}=\overset{N}{\underset{i=0}{\sum }}{\bar{X_{i}}}^{2}\cdot AUC\left(\bar{X_{i}}\right)$$
where $AUC\left(\bar{X_{i}}\right)$ is the AUC value that was calculated after the ROC curve analysis of variable $\bar{X_{i}}$ for groups $G_{A}$ and $G_{B}$.

Approach D: Sum of squared scores, scaled by AUC squared

Since AUC is bounded in the $\left[0,\;1\right]$ range and values near 1 indicate a better discriminant potential, it makes sense to raise the AUC value of each variable to the square power, further boosting the variables with high AUCs (similarly to approach B). Therefore, the final score is
$$S_{\Sigma^{2}AUC^{2}}=\overset{N}{\underset{i=0}{\sum }}{\bar{X_{i}}}^{2}\cdot AUC{\left(\bar{X_{i}}\right)}^{2}$$

Approach E: Scaling the normalized variables with standard deviation

Finally, another approach that has better discrimination potential is to use the standard deviation of each normalized variable. As we know, each normalized variable is bounded to the $\left[0,\;1\right]$ range and, due to the min-max normalization, there is at least one instance of 0 and at least one instance of 1. Ideally, a variable that could perfectly discriminate our two groups $G_{A}$ and $G_{B}$ would contain two subsets of values, one near 0 and one near 1, allowing for a perfect linear separation. On the contrary, if a variable has a uniform density of values throughout the $\left[0,\;1\right]$ range, a linear separation (even though possible) is not very probable, since individuals from the two groups can have similar scores. This concept can be measured by standard deviation, since in the first case the standard deviation will be low whereas in the second the standard deviation will be high. Therefore, if we want to further boost the variables with values that are gathered instead of spread, we can divide each variable by their standard deviation for all of the above approaches, resulting in four more possible total scores:$$S_{\Sigma }^{\sigma }=\overset{N}{\underset{i=0}{\sum }}\frac{\bar{X_{i}}}{\sigma_{X_{i}}}$$
$$S_{\Sigma^{2}}^{\sigma }=\overset{N}{\underset{i=0}{\sum }}\frac{{\bar{X_{i}}}^{2}}{\sigma_{X_{i}}}$$
$$S_{\Sigma^{2}AUC}^{\sigma }=\overset{N}{\underset{i=0}{\sum }}\frac{{\bar{X_{i}}}^{2}}{\sigma_{X_{i}}}\cdot AUC\left(\bar{X_{i}}\right)$$
$$S_{\Sigma^{2}AUC^{2}}^{\sigma }=\overset{N}{\underset{i=0}{\sum }}\frac{{\bar{X_{i}}}^{2}}{\sigma_{X_{i}}}\cdot AUC{\left(\bar{X_{i}}\right)}^{2}$$

#### 4.2.2. SCD vs. HC Analysis

There were five (5) scores on which only direct effects of the diagnostic group were found with regard to the SCD and HC subsamples. There comprised the WMCUT S3, ICT-RST 1 & 2, ICT-RST FS, ICT-RST 3, and ICT-RST 4b. As evident in the next table, the total score formula that achieved a better Sensitivity/Specificity combination was the $S_{\Sigma^{2}}^{\sigma }$. The $S_{\Sigma^{2}}^{\sigma }$ formula comprising the subtasks’ scores mentioned above resulted in excellent sensitivity (100%) and specificity (95%) to discriminate SCD from HC (AUC = 0.97, 95% CI = 0.9–1.00, *p* < 0.001) with a cutoff of 1.2236 (Table 5 and Table 6, Figure 4).

#### 4.2.3. SCD vs. MCI Analysis

There were four (4) scores on which only direct effects of the diagnostic group were found with regard to the SCD and MCI subsamples. These were ICT-RST 1 & 2, ICT-RST SW, ICT-RST 4a, and WINDOWS A. As evident in the next table, the total score formula that achieved a better Sensitivity/Specificity combination is $S_{\Sigma^{2}AUC^{2}}^{\sigma }$ which resulted in good sensitivity (90.3%) and specificity (82.8%) to discriminate SCD from MCI (AUC = 0.89 95% CI = 0.8–0.9, *p* < 0.001) with a cutoff of 8.4165 (Table 7 and Table 8, Figure 5).

#### 4.2.4. HC vs. MCI Analysis

There were five (5) scores on which only direct effects of the diagnostic group were found regarding HC and MCI subsamples: ICT-RST 1 & 2, ICT-RST 4a, ICT-RST 4b, WINDOWS A, and WINDOWS B. As evident in the next table, the total score formula that achieved a better Sensitivity/Specificity combination is $S_{\Sigma^{2}AUC^{2}}$. The $S_{\Sigma^{2}AUC^{2}}$ formula resulted in excellent sensitivity (100%) and specificity (90%) to discriminate HC from MCI (AUC = 0.97, 95% CI = 0.9–1.00, *p* < 0.001) with a cutoff of 0.4804 (Table 9 and Table 10, Figure 6).

## 5. Discussion

The aim of the present study was to extend the existing R4Alz battery with extra tasks comprising further cognitive control tasks, (plus a Visual Fluency Task) as well as episodic memory tasks, in order to raise the discrimination ability of the original R4Alz battery between SCD and MCI, but also to enhance the already excellent discriminant validity between SCD and HC. In addition, the present study studies the discrimination between MCI and HC, an analysis that was not performed in the last published paper on the R4Alz battery [9].

### 5.1. Age and Educational Level Effects

Regarding the first hypothesis of the study, namely that the battery’s subtasks would not be affected by demographic variables such as age or education, our results showed that there was not any significant (*p* < 0.004) mediation of age and education in any of the subtasks and tasks of the R4Alz-R battery between all groups. Regarding education, and according to the former validation study of the original R4Alz battery [9], the tasks and the subtasks of the battery were also free of education effects, a finding that is in agreement with the current study. This finding is important since tests that are educational-level-free may ensure, to an extent, the diagnostic validity of the test and as a consequence the accuracy of the categorization of the neurodegeneration stages. To be more precise, nowadays several neuropsychological tests exist, mainly assessing cognitive control, such as TMT and STROOP, which are affected by demographic characteristics such as education [53,54,55,56]. High educational levels can lead to misdiagnosis in neurodegeneration, according to the theory of cognitive reserve [57]. As a result, people with a high educational level can be under-diagnosed, and people with a low educational level can be falsely categorized as dementia patients. Therefore, in order to avoid demographic biases, researchers usually propose to adjust neuropsychological tests according to education [8,58]. Since R4Alz-R seems to not be affected by education, it could be a more valid diagnostic tool, free of biases caused by demographics.

Regarding age, there are studies indicating that there are age-related changes especially in tasks that require a high degree of cognitive control abilities (inhibition, updating, and shifting), [59,60,61,62]. These findings are based on the prefrontal-executive theoretical model, that supports the idea that frontal lobes, important cerebral structures for executive processes, are the most vulnerable to the advancing of age [15,21]. In our study, it seems that age does not mediate diagnostic group effects on the sample’s performance on the R4Alz tasks and subtasks, despite the fact that the three groups differed at baseline according to age, with HC being younger than MCI. A possible reason for these findings is that the R4Alz battery was initially designed to not be affected by age; in the original R4Alz battery, in order to avoid any age effect in tasks and subtasks, we have increased the time required for each task completion, depending on the idea that age effects in the test’s performance would disappear if components such as motor-speed would be accounted for [63,64,65]. Moreover, according to the Temporal Hypothesis for Compensation [63], there is a compensatory mechanism characterized by age-related delayed cerebral activation, allowing for cognitive performance to be preserved at the expense of speed processing. Therefore, based on the above hypothesis, we considered that if we raise the time needed for the completion of each task, healthy older adults would probably perform well, while a diminished performance would probably reflect cognitive changes due to a neurocognitive disorder, rather than an age-related difficulty. In other words, if we give more time to an older adult person to interpret a task, she/he will probably manage to do it, if she/he is cognitively healthy. Therefore, consistent with all of the above, the lack of any age effects on the R4Alz-R tasks is probably associated on the one hand with the design of the battery, and on the other hand with the fact that the main effects of diagnosis on performance were in fact the “real” biological factors that underlie age and education effects on the performance of older adults to a large extent [49]. To conclude, regarding the demographic’s effects in the battery’s performance, it seems that neither age nor education has a significant effect on performance, and therefore the battery seems free of biases that may affect its validity.

### 5.2. R4Alz-R’s Differential Capacity

The main target of the current study was to enhance the validity of the original R4Alz battery to discriminate SCD from HC and MCI, via the addition of episodic memory and cognitive flexibility tasks. According to our results, different cognitive tasks and subtasks of the R4Alz-R battery differentiated HC from SCD, SCD from MCI, and finally SCD from MCI. The reason for this diversity may reflect the non-uniform nature of the deficits that people with SCD [66] and people with MCI face [67].

### 5.3. Differential Capacity between HC and SCD

Regarding the differential ability of SCD from HC, according to our results, the tasks of (a) updating of working memory (WMCUT S3), (b) inhibition and switching (ICT-RCT S1 & S2 and ICT-RCT FS), as well as (c) and (d) the two cognitive flexibility tasks (CFT and CFT Con. B) were able to differentiate healthy people from people experiencing subjective cognitive decline. The above results are in agreement with the preceding R4Alz’s validation study, in which tasks demanding updating, inhibition, and the combination of inhibition and switching of attention (cognitive flexibility) functions differentiated these groups. Therefore, the addition of the new cognitive flexibility task and the extra levels of difficulty strengthened the discriminating ability of the battery regarding the difference between SCD and HC.

In the present study, despite the fact that there was a strong capacity of the above cognitive control abilities to differentiate SCD from HC, a significant differential ability of episodic memory tasks was not observed. Nowadays, there are several research papers which utilize either biomarkers or neurocognitive data, that indicate that people with SCD show an executive function decline, rather than a memory decline [12,68,69], a fact that supports the idea that cognitive changes may start from other cognitive areas than the hippocampus and parahippocampial areas, such as the frontal lobe [70,71] and prefrontal cortex [72], which are areas closely related to executive function [73]. Notably, Ohlhauser, Parker, Smart, Gawryluk, and the Alzheimer’s Disease Neuroimaging Initiative in 2019 [74] attempted to examine whether differences in white matter integrity between individuals with SCD (n = 30) and healthy controls (n = 44) exist and how the white matter integrity is related to memory and executive function. According to their results, there is a correlation between executive function and white matter integrity. Specifically, lower white matter integrity was noticed in widespread regions such as bilateral corticospinal tracts, superior and inferior longitudinal fasciculi, fronto-occipital fasciculi, corpus callosum, forceps major and minor, hippocampi, anterior thalamic radiations, and the cerebellum. Moreover, lower white matter integrity was also related to lower executive function in individuals with SCD compared to HC [74]. In other words, the above study supports the idea that there are microstructural differences in white matter between the SCD and HC, which may possibly be related to executive function. According to the authors, people with SCD may be able to functionally compensate for structural changes, until frontal degeneration is conspicuous, and cognitive decline becomes measurable [74].

In agreement with the previous studies, there are also studies indicating that the nature of cognitive complaints is more related to difficulties in attentional and cognitive control than pure memory complaints [75]. Valech et al. in 2017 assessed healthy controls and people with pre-AD (people who fulfilled the criteria for SCD and had abnormal levels of Aβ). The aim of their study was to investigate specific characteristics of SCD in pre-AD, by utilizing the Subjective Cognitive Decline Questionnaire (SCD-Q). Regarding the executive function domains, they have found significantly higher complaints in executive tasks in the pre-AD group and specifically in areas which were related to difficulties in (a) concentration, (b) using electronic devices, (c) starting a conversation, and (d) multi-tasking. According to their explanation, the decline in executive function may be simultaneously presented with memory complaints; however, “executive complaints appear to be more specific to AD pathophysiology given the cofounding effect of normal aging in memory complaints”. In our study, a questionnaire which would assess the specific cognitive complaints of SCD was not used, since there are studies that indicate that the currently available SCD questionnaires lack content validity evaluation [76]. Nevertheless, the sample of people with SCD have mentioned problems in executive function during their clinical interview. Despite the fact that our SCD sample was not checked for AD pathophysiology, according to our results, executive function abilities seem to be quantitatively (thus objectively) different between SCD and HC, with HC outperforming people with SCD.

As for the discriminant potential of the R4Alz-R battery between SCD and HC, in order to derive a cut-off point, we used the aforementioned tasks that showed significant differences in discrimination. The best discrimination was derived from the sum of squared scores normalized with standard deviation total score ($S_{\Sigma^{2}}^{\sigma }$), having a cutoff score of 1.22, resulting in an excellent sensitivity and specificity (100% and 95%, respectively).

Compared to the former validation study of the R4Alz in which the dissemination potential was also high (AUC 0.960, Sens 85.3%, and Spec. 92.9%), and including the same cognitive control abilities, it seems that the addition of extra levels of difficulty, especially in the domain of cognitive flexibility, has resulted in the enhancement of the discriminant validity of the battery. Regarding the CFT2 condition, it requires the examinee to internally remember the basic rule (which is to shift attention in each set by following the sequence of the colours that she/he has to deactivate) and also to inhibit the irrelevant stimuli (for example, in the first set, the first green pad, and afterwards, the next red pad, and again, the next green pad), whilst in the next set, the examinee has to also externally shift his attention by deactivating both green and red pads, however starting with a different color and following the opposite direction from the previous set. Obviously, the level of difficulty for this condition is too high by simultaneously requiring an internal shifting between colours in each set and an external shifting of direction between sets. Therefore, as the memory load is too high, the participant has to design an effective strategy in order to remember the instructions/rules and correctly perform the switches. Furthermore, it is worth noting that in the specific condition it seems that another cognitive ability is also required, this being the updating of working memory, since the examinee has to update which was the starting color and the deactivation direction of the previous set, in order to shift attention. There is a possibility that the cognitive flexibility tasks plus updating play a crucial role in the differentiation of SCD from healthy controls. Furthermore, the study of Smart & Krawitz in 2015 [77] assessed the impact of the Iowa Gambling Task (IGT) for detecting measurable cognitive differences in SCD compared to healthy older adults. In their study, they indicate that there are measurable differences in risky decision making in older adults with SCD, compared to controls [77]. The interpretation of their results is quite interesting, since they suggest that people with SCD score worse in the IGT because they have difficulties in updating in environments of uncertainty and doubt. In other words, according to the authors, when the environment is unstable and changing rapidly, people with SCD “may be setting their learning rate differently because of impairment in their ability to track volatility” (p. 983).

The discrimination potential of the R4Alz-R to differentiate SDC from HC with exact scores and specific cutoffs is the most important result of the study. Despite the fact that there are also other studies which differentiate SCD from HC with a high accuracy, the discriminant validity of R4Alz-R is superior since it is excellent (ROC-based). Indicatively, the study of Lazarou at al. in 2021 [7] showed that the Memory Alteration Test (M@T) had a fair ability for differentiation between SCD and older healthy controls with 81% sensitivity and 61% specificity (AUC = 0.76) [7]. The above result was expected since M@T is a verbal episodic memory test, and thus it does not assess abilities of cognitive control that seem to be affected at an early stage of the neurodegeneration [11,12]. Therefore, we consider that the impact of the R4Alz-R is crucial for the objective neuropsychological detection of the SCD status, which seems to not be “subjective” but to represent objective neurodegeneration.

### 5.4. Differential Capacity between SCD and MCI

With regard to the ability of the R4Alz-R battery to differentiate SCD from MCI, performances on four tasks were shown to be able to significantly differentiate SDC from MCI people. These tasks are (a) Inhibitory Control & Task/Rule Switching Task Subtasks 1 (Inhibition) and 2 (Task/Rule switching) (ICT-RST 1 & 2), (b) Switch Errors on the Inhibitory Control & Task/Rule Switching Task (ICT-RST 1 & 2 SE), (c) 1st level of difficulty of the Cognitive Flexibility Task (CFT Condition a), and (d) the 1st level of difficulty of the Episodic Memory Task of Windows (EMT-W Condition a).

Τhe most interesting result of the current analysis concerns the new windows task. It has been designed to assess visual episodic memory and to examine whether such a task could offer more in the differentiation of SCD from MCI. According to the results, the windows task showed a significant difference between SCD and MCI. The first level of difficulty of the Windows task showed a statistically significant difference between groups. According to literature, deficits in visual episodic memory not only exist in people with MCI, but they are also considered to be an indicator of conversion in major neurocognitive diseases [78]. Despite the fact that episodic memory is impaired in MCI, in SCD, episodic memory seems to be intact. For example, the study of Viviano, & Damoiseaux, in 2021 (p. 12) [79], shows that the degree of SCD does not predict baseline cognitive performance nor longitudinal change in visual working memory or episodic memory.

Hence, based on our results, SCD differs from HC in terms of executive function, whereas SCD differs from MCI both in terms of executive function and episodic memory. Evidently, even though the course of neurodegeneration might begin with cognitive changes in executive function abilities, when the deficits are measurable and objective as occurs in amnestic MCI, the memory problems seem to be more prominent [80,81].

Overall, the extra tests that were added to the initial battery seem to have played a crucial role in the groups’ differentiation. Apart from the windows episodic memory task which showed a statistical significance, the cognitive flexibility task which comprised a combination of inhibition and switching also presented a good differentiation. In the current study, the first level of difficulty of the cognitive flexibility task requires the examinee not only to inhibit the irrelevant stimuli and switch the attention from the red pads to green, but also to switch the attention concerning the direction of deactivations (from right to left). Obviously, this task requires a raised mnemonic load, since the participants have to maintain the rule in their memory and to appropriately react based on the stimuli. Cognitive flexibility and inhibitory control are usually utilized in the literature for the differential diagnosis of MCI from cognitive healthy older adults with great success [82]. According to a recent review regarding whether inhibitory and interference control, conflict control, and cognitive flexibility could provide a better diagnosis of MCI, these abilities are useful in discrimination, suggesting their systematic utilization in neuropsychological batteries [83].

As far as the discriminant validity of the R4Alz-R is concerned, it showed an excellent classification between SCD and MCI, reaching a sensitivity of 90.3% and a specificity of 82.8%. To our knowledge, there are only a few batteries which attempt to differentiate SCD from MCI by providing objective cutoff scores. For example, the Montreal Cognitive Assessment Scale (MoCA) seems to be affected by educational level and to have a fair to good discrimination ability for differentiating SCD from MCI, with a sensitivity between 71–77% and specificity from 67% to 84.2%, for low [6], middle, and high educational level, whilst the Memory Alteration Test (M@T), which is also used for the same differentiation, shows a poor discrimination ability (63% sensitivity and 73% specificity) [7].

### 5.5. Differential Capacity between HC and MCI

According to the results, all the tasks of the R4Alz-R battery achieved statistically significant differentiation between HC and MCI. That typically means that healthy adults of over 50 years of age have different performance than people with MCI in terms of: (a) working memory (short-term store, central executive, updating), (b) attentional control, (c) inhibitory control and switching, (d) cognitive flexibility, and (e) episodic memory capacity.

The vast majority of people with MCI that participated in our sample had the same subtype of MCI, which was multiple-domain amnestic MCI. This specific MCI subtype is considered to have the poorest performance among the four subgroups of MCI [84]. This type is characterized by clear deficits in episodic and working memory, attention, and executive function and is also recognizable via several neuropsychological tests [84]. Moreover, more complex abilities such as switching, planning, and verbal fluency seem to be the executive functions that distinguish between the people who are in danger of conversion into dementia and the people with MCI who will remain stable [85]. Generally, there are also other batteries, computerized or not, which assess general cognitive abilities such as RBANS [86] or BrainCheck [87] and can accurately differentiate healthy adults from people with MCI (sensitivity ranging from 77% to 87% and specificity 77% to 81%). The fact that all tasks of the R4Alz-R showed a better differentiating accuracy between HC and MCI (100% sensitivity and 90% specificity) than the studies mentioned before indicates that the battery can distinguish the healthy status from the pathological quite reliably.

### 5.6. Comments on the Total Score Creation Process

At this point, it is worth making some comments regarding the R4Alz’s-R total score creation process. As evident from the presented analyses, the proposed total score formulations achieved a better differentiating capacity in comparison to the simple sum of all scores, which is usually utilized in such cases. This makes sense, since all the proposed formulae, except from the fact that they offer a non-linear way of combining the individual scores, are scaled with coefficients that objectively measure the discriminant capacity of each individual sub-score. Nevertheless, as evident from the results, not one single total score suggestion was presented as dominant (e.g., in the SCD vs. HC, $S_{\Sigma^{2}}^{\sigma }$ gave the best result, whereas in the MCI vs. SCD case, the $S_{\Sigma^{2}AUC^{2}}^{\sigma }$ was optimal). This fact indicates that the researchers who would like to follow this procedure, should compute all eight total score formulae and pick the best for their needs, according to the results.

### 5.7. Limitations and Future Work

Even though the results indicate that R4Alz-R is a good tool for the differential diagnosis of the different stages of cognitive decline, the study has several limitations that must be addressed. First, there is a limited number of persons participating in each group, a fact that does not allow us to generalize our results to the public, and therefore future studies with larger samples are needed. Secondly, in this study, we did not assess any correlation between biological indicators such as APOE status and performance in the R4Alz-R battery. It would be interesting to categorize the SCD and HC participants as APOE4 and no-APOE4 carriers and therefore to split our sample based on who is at biological risk of developing Alzheimer’s, afterwards assessing for performance differences. Furthermore, other biomarkers of Alzheimer’s disease were also not available in our study and especially amyloid and tau. This strand of research may strengthen our results and is definitely one of our next targets. Therefore, for further studies, it is worth exploring the discrimination ability of the R4Alz-R battery, in the biological spectrum of Alzheimer’s Disease (SCD and MCI with biological AD, and SCD and MCI with non-AD). Finally, one of our next goals is to categorize our sample in more homogeneous groups such as early MCI and late MCI, or according to their subtypes, to investigate whether the subgroup/subtype of MCI plays a role in the discriminate validity of the R4Alz-R.

## 6. Conclusions

The aim of the present study was the enhancement of the R4Alz battery with extra tasks to assess different levels of cognitive control and episodic memory, to increase its ability to differentiate SCD from HC over 50 years of age, and MCI. According to the results, the R4Alz-R has a perfect discriminant potential to differentiate people with SCD from HC as well as HC from MCI, and an excellent discriminant potential between SCD and MCI. Components of cognitive control such as updating, inhibition, switching, and cognitive flexibility are the abilities that differentiated SCD from HC, indicating that the executive function difficulties are presented quite early in the course of the neurodegenerating diseases. On the other hand, episodic memory plus cognitive control (inhibition, switching, and cognitive flexibility) can differentiate SCD from MCI, indicating that, as the cognitive decline progresses, the episodic memory problems are more prominent along with executive dysfunction. According to our results, the SCD seems to be a separate diagnostic category which differs quantitatively from healthy adults and people with MCI, as we provided objective, measurable cutoff scores. The above findings enhance the hypothesis that people with SCD comprise a group at risk, and therefore, specific preventing strategies have to be implemented. R4Alz-R is considered to be a very useful battery for the diagnosis of the different stages of incipient cognitive decline, and a tool that is free of age or education effects. Furthermore, additionally to the fact that the R4Alz-R battery is not affected by age and education, it is also a battery which mainly utilizes visual–spatial and audio-related abilities such as colours, photos of windows or animals, and basic sounds; therefore, we consider that it is a battery also free of cultural effects. Based on all of the above, we consider that the R4Alz battery is a cross-cultural tool that would be quite useful for utilization in population-based studies on SCD.

## Figures and Tables

**Figure 1 diagnostics-13-00338-f001:**
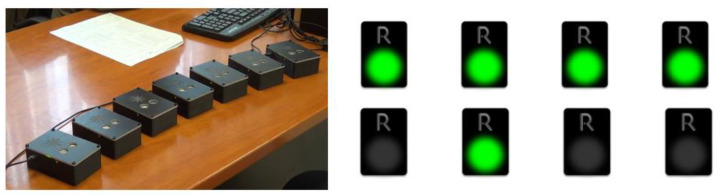
The R4Alz original version and the R4Alz-R (electronic form).

**Figure 2 diagnostics-13-00338-f002:**
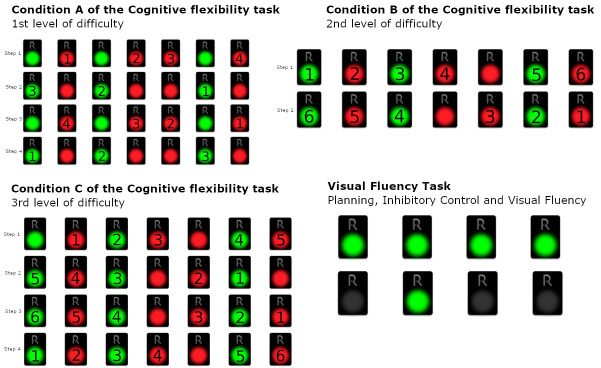
Condition A–C of the Cognitive Flexibility Task—1st, 2nd, and 3rd levels of difficulty and Visual Fluency Task (Planning, Inhibitory Control, and Visual Fluency).

**Figure 3 diagnostics-13-00338-f003:**
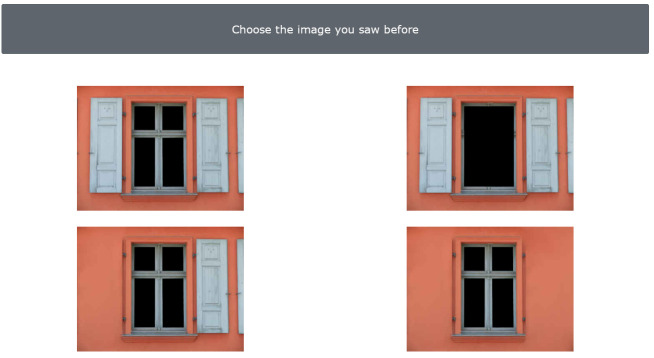
Episodic Memory Task—Windows (1st level of difficulty)–Caption in Greek: Select the image you saw before.

**Figure 4 diagnostics-13-00338-f004:**
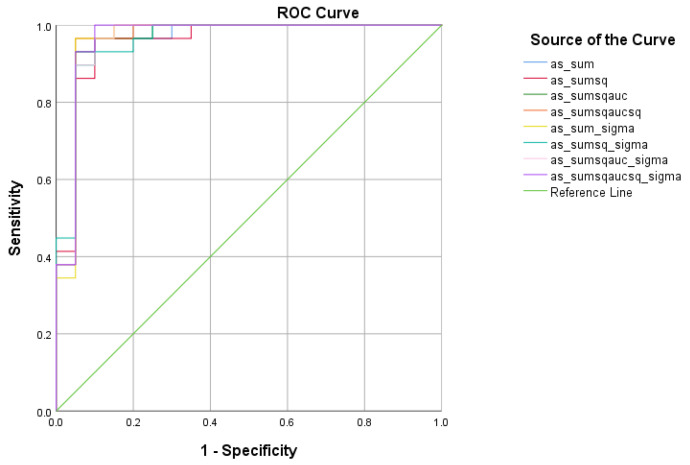
ROC curve analysis of the 8 formulae that better discriminate between Subjective Cognitive Decline (SCD) and Cognitively Healthy Advanced-age Adults (HC).

**Figure 5 diagnostics-13-00338-f005:**
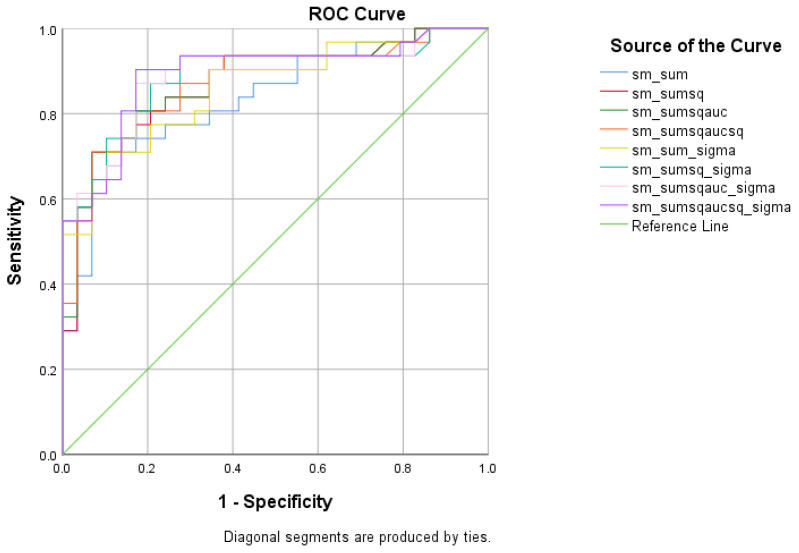
ROC curve analysis of the 8 formulae that better discriminate between Subjective Cognitive Decline SCD and Mild Cognitive Impairment (MCI).

**Figure 6 diagnostics-13-00338-f006:**
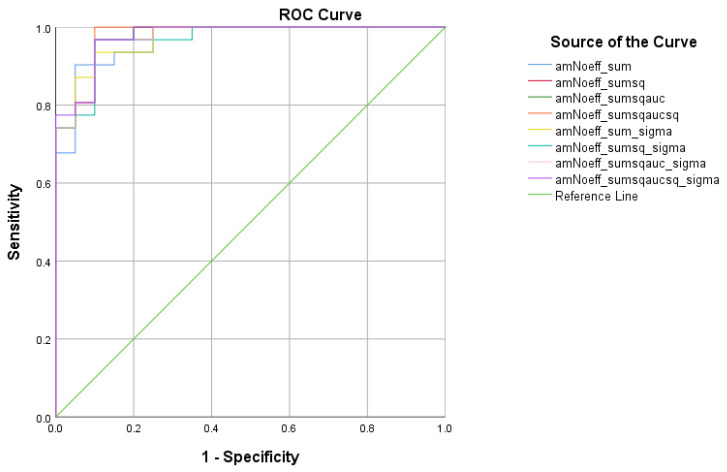
ROC curve analysis of the 8 formulae that better discriminate between Cognitively Healthy Advanced-age Adults (HC) and Mild Cognitive Impairment (MCI).

**Table 1 diagnostics-13-00338-t001:** Demographic characteristics of the participants of the study (n = 80).

Diagnostic Groups				
Characteristics	HC (n =20)	SCD (n = 29)	MCI (n = 31)	*p*
Age M (SD)	61.60 (6.58)	61.17 (6.58)	68.67 (8.43)	<0.05
Gender (Male/Female)	6 M/14 F	9 M/20 F	7 M/24 F	>0.05
Education M (SD)	16.30 (3.04)	13.31 (4.20)	13.45 (4.31)	<0.05
MoCA M (SD)	28.29 (1.35)	27.17 (3.60)	25.14 (1.88)	<0.05

Abbreviations: HC: Cognitively Healthy Advanced-age Adults; SCD: Adults with Subjective Cognitive Decline; MCI: People with Mild Cognitive Impairment, MoCA: Montreal Cognitive Assessment.

**Table 2 diagnostics-13-00338-t002:** (**a**) Direct effects of diagnosis on the R4Alz-R tasks’ and subtasks’ performance according to Mediation Analysis, between the groups of SCD and HC (** p* < 0.004). (**b**) Indirect effects of diagnosis—via age and education—on the R4Alz-R tasks’ and subtasks’ performance according to Mediation Analysis, between the groups of HC and SCD (* *p* < 0.004).

(a)										
Direct Effects									95% Confidence Interval	
					b	SE	z-Value	*p*	Lower	Upper
diagnosis	→	WMCUT S 1			0.039	0.299	0.131	0.896	−0.545	0.771
diagnosis	→	WMCUT S 2			−0.682	0.286	−2.383	0.017	−1.310	−0.028
diagnosis	→	WMCUT S 3			−1.999	0.665	−3.005	**0.003 ***	−3.025	−1.123
diagnosis	→	ACT			4.242	1.726	2.459	0.014	1.333	7.329
diagnosis	→	ICT−RST 1 & 2			7.270	2.197	3.309	**<0.001 ***	3.626	10.540
diagnosis	→	ICT−RST 1 & 2 SE			0.437	0.200	2.178	0.029	0.057	0.772
diagnosis	→	ICT−RST 1 & 2 FS			1.188	0.398	2.983	**0.003 ***	0.390	1.851
diagnosis	→	CFT			3.735	0.849	4.398	**<0.001 ***	1.761	5.522
diagnosis	→	CFT Con. a			0.371	0.516	0.718	0.473	−0.591	1.094
diagnosis	→	CFT Con. b			1.758	0.559	3.145	**0.002 ***	0.610	2.876
diagnosis	→	CFT Con. c			1.262	0.644	1.961	0.050	−0.176	2.681
diagnosis	→	VFT			−1.653	0.624	−2.649	0.008	−3.052	−0.551
diagnosis	→	EMT−W Con. a			0.838	0.800	1.048	0.294	−0.868	2.509
diagnosis	→	EMT−W Con. b			1.203	0.598	2.014	0.044	−0.114	2.557
**(b)**										
**Indirect Effects**									**95% Confidence Interval**	
					**b**	**SE**	**z-Value**	** *p* **	**Lower**	**Upper**
diagnosis	→	Age	→	WMCUT S 1	−0.402	0.177	−2.269	0.023	−0.869	−0.177
diagnosis	→	Education	→	WMCUT S 1	−0.053	0.107	−0.498	0.619	−0.369	0.083
diagnosis	→	Age	→	WMCUT S 2	−0.297	0.147	−2.028	0.043	−0.641	−0.095
diagnosis	→	Education	→	WMCUT S 2	−0.172	0.118	−1.459	0.144	−0.588	−0.007
diagnosis	→	Age	→	WMCUT S 3	−0.268	0.258	−1.037	0.300	−0.880	0.034
diagnosis	→	Education	→	WMCUT S 3	−0.028	0.234	−0.121	0.904	−0.650	0.439
diagnosis	→	Age	→	ACT	1.240	0.759	1.633	0.102	−0.035	3.383
diagnosis	→	Education	→	ACT	0.274	0.614	0.447	0.655	−1.108	1.627
diagnosis	→	Age	→	ICT−RST 1 & 2	0.906	0.855	1.060	0.289	−0.427	4.029
diagnosis	→	Education	→	ICT−RST 1 & 2	1.816	1.011	1.797	0.072	0.330	4.619
diagnosis	→	Age	→	ICT−RST 1 & 2 SE	0.022	0.073	0.298	0.766	−0.139	0.248
diagnosis	→	Education	→	ICT−RST 1 & 2 SE	0.200	0.101	1.990	0.047	0.047	0.489
diagnosis	→	Age	→	ICT−RST 1 & 2 FS	0.125	0.150	0.828	0.408	−0.137	0.585
diagnosis	→	Education	→	ICT−RST 1 & 2 FS	0.275	0.171	1.607	0.108	0.066	0.705
diagnosis	→	Age	→	CFT	0.583	0.368	1.582	0.114	−0.067	1.718
diagnosis	→	Education	→	CFT	0.268	0.313	0.855	0.393	−0.129	1.244
diagnosis	→	Age	→	CFT2 Con. a	0.106	0.190	0.555	0.579	−0.227	0.537
diagnosis	→	Education	→	CFT2 Con. a	0.341	0.219	1.558	0.119	0.008	1.069
diagnosis	→	Age	→	CFT2 Con. b	0.128	0.207	0.621	0.535	−0.226	0.530
diagnosis	→	Education	→	CFT2 Con. b	0.364	0.236	1.543	0.123	0.021	1.044
diagnosis	→	Age	→	CFT2 Con. c	0.418	0.275	1.523	0.128	−0.031	1.228
diagnosis	→	Education	→	CFT2 Con. c	0.418	0.271	1.539	0.124	0.035	1.115
diagnosis	→	Age	→	VFT	−0.573	0.302	−1.901	0.057	−1.376	−0.051
diagnosis	→	Education	→	VFT	−0.240	0.235	−1.018	0.309	−1.088	0.126
diagnosis	→	Age	→	EMT−W Con. a	0.228	0.300	0.760	0.447	−0.203	0.931
diagnosis	→	Education	→	EMT−W Con. a	0.186	0.289	0.643	0.520	−0.189	1.115
diagnosis	→	Age	→	EMT−W Con. b	0.350	0.248	1.409	0.159	−0.098	1.012
diagnosis	→	Education	→	EMT−W Con. b	0.188	0.220	0.855	0.393	−0.245	0.622

Abbreviations: WMCUT S1: Working Memory Capacity and Updating Task—working memory component of short-term store; WMCUT S2: Working Memory Capacity and Updating Task—working memory component of central executive; WMCUT S3: Working Memory Capacity and Updating Task—working memory updating & working memory component of the episodic buffer; ACT: Attentional Control Task; ICT-RST 1 & 2: Inhibitory Control & Task/Rule Switching Task Subtasks 1 (Inhibition) and 2 (Task/Rule switching); ICT-RCT SE: Total number of errors of Inhibitory Control & Task/Rule Switching Task; ICT-RST FS: Failed sets on the Inhibitory Control & Task/Rule Switching Task; CFT: Cognitive Flexibility Task; CFT Con. a: Cognitive Flexibility Task—1st level of difficulty; CFT Con. b: Cognitive Flexibility Task—2nd level of difficulty; CFT Condition c: Cognitive Flexibility Task—3rd level of difficulty; VFT: Visual Fluency Task (Planning, Inhibitory Control, and Visual Fluency); EMT-W Con. a; Episodic Memory Task—Windows (1st level of difficulty); EMT-W Con. b; Episodic Memory Task—Windows (2nd level of difficulty).

**Table 3 diagnostics-13-00338-t003:** (**a**) Direct effects of diagnosis on the R4Alz-R tasks and subtasks’ performance according to Mediation Analysis, between the groups of SCD and MCI (* *p* < 0.004) (**b**) Indirect effects of diagnosis—via age and education—on the R4Alz-R tasks’ and subtasks’ performance according to Mediation Analysis, between the groups of SCD and MCI (* *p* < 0.004).

(a)										
Direct Effects									95% Confidence Interval	
					b	SE	z-Value	*p*	Lower	Upper
diagnosis	→	WMCUT S 1			−0.253	0.204	−1.236	0.217	−0.698	0.164
diagnosis	→	WMCUT S 2			−0.483	0.211	−2.292	0.022	−0.932	−0.061
diagnosis	→	WMCUT S 3			−1.040	0.655	−1.587	0.112	−2.288	0.415
diagnosis	→	ACT			5.439	1.950	2.789	0.005	1.813	9.321
diagnosis	→	ICT-RST 1 & 2			9.710	2.690	3.610	**<0.001 ***	4.057	14.934
diagnosis	→	ICT-RST 1 & 2 SE			1.153	0.257	4.485	**<0.001 ***	0.678	1.671
diagnosis	→	ICT-RST 1 & 2 FS			1.078	0.406	2.655	0.008	0.219	1.848
diagnosis	→	CFT			2.099	0.794	2.643	0.008	0.588	3.757
diagnosis	→	CFT2 Con. a			1.597	0.498	3.204	**0.001 ***	0.551	2.546
diagnosis	→	CFT2 Con. b			1.066	0.467	2.284	0.022	−0.041	2.007
diagnosis	→	CFT2 Con. c			0.561	0.446	1.257	0.209	−0.374	1.471
diagnosis	→	VFT			−0.945	0.458	−2.064	0.039	−1.883	−0.017
diagnosis	→	EMT-W Con. a			2.374	0.558	4.257	**<0.001 ***	1.222	3.419
diagnosis	→	EMT-W Con. b			0.564	0.519	1.087	0.277	−0.507	1.496
**(b)**										
									**95% Confidence Interval**	
**Indirect Effects**					**b**	**SE**	**z-Value**	** *P* **	**Lower**	**Upper**
diagnosis	→	Age	→	WMCUT S 1	−0.072	0.097	−0.745	0.456	−0.301	0.124
diagnosis	→	Education	→	WMCUT S 1	2.377	0.004	0.061	0.952	−0.044	0.072
diagnosis	→	Age	→	WMCUT S 2	−0.063	0.085	−0.739	0.460	−0.259	0.103
diagnosis	→	Education	→	WMCUT S 2	0.001	0.009	0.120	0.905	−0.051	0.081
diagnosis	→	Age	→	WMCUT S 3	−0.174	0.238	−0.733	0.463	−0.816	0.245
diagnosis	→	Education	→	WMCUT S 3	0.011	0.085	0.129	0.897	−0.193	0.337
diagnosis	→	Age	→	ACT	0.645	0.868	0.743	0.457	−1.066	2.767
diagnosis	→	Education	→	ACT	−0.033	0.257	−0.130	0.897	−1.153	0.517
diagnosis	→	Age	→	ICT−RST 1 & 2	0.664	0.912	0.729	0.466	−1.081	3.191
diagnosis	→	Education	→	ICT−RST 1 & 2	−0.067	0.515	−0.130	0.896	−1.757	1.132
diagnosis	→	Age	→	ICT−RST 1 & 2 SE	0.041	0.059	0.689	0.491	−0.061	0.324
diagnosis	→	Education	→	ICT−RST 1 & 2 SE	−0.017	0.133	−0.131	0.896	−0.304	0.274
diagnosis	→	Age	→	ICT−RST 1 & 2 FS	0.090	0.124	0.721	0.471	−0.128	0.511
diagnosis	→	Education	→	ICT−RST 1 & 2 FS	−0.016	0.121	−0.130	0.896	−0.273	0.271
diagnosis	→	Age	→	CFT	0.269	0.362	0.744	0.457	−0.490	1.101
diagnosis	→	Education	→	CFT	−0.019	0.145	−0.130	0.897	−0.546	0.280
diagnosis	→	Age	→	CFT2 Con. a	0.035	0.067	0.520	0.603	−0.083	0.435
diagnosis	→	Education	→	CFT2 Con. a	−0.020	0.157	−0.130	0.896	−0.428	0.314
diagnosis	→	Age	→	CFT2 Con. b	0.020	0.053	0.376	0.707	−0.074	0.394
diagnosis	→	Education	→	CFT2 Con. b	−0.022	0.165	−0.130	0.896	−0.365	0.356
diagnosis	→	Age	→	CFT2 Con. c	0.064	0.095	0.675	0.500	−0.068	0.472
diagnosis	→	Education	→	CFT2 Con. c	−0.009	0.068	−0.130	0.897	−0.233	0.148
diagnosis	→	Age	→	VFT	−0.068	0.100	−0.680	0.497	−0.476	0.095
diagnosis	→	Education	→	VFT	0.011	0.085	0.130	0.897	−0.155	0.301
diagnosis	→	Age	→	EMT−W Con. a	0.007	0.056	0.131	0.896	−0.112	0.275
diagnosis	→	Education	→	EMT−W Con. a	0.002	0.020	0.115	0.908	−0.102	0.215
diagnosis	→	Age	→	EMT−W Con. b	0.004	0.052	0.069	0.945	−0.195	0.259
diagnosis	→	Education	→	EMT−W Con. b	−0.003	0.024	−0.122	0.903	−0.215	0.153

Abbreviations: WMCUT S1: Working Memory Capacity and Updating Task—working memory component of short-term store; WMCUT S2: Working Memory Capacity and Updating Task—working memory component of central executive; WMCUT S3: Working Memory Capacity and Updating Task—working memory updating & working memory component of the episodic buffer; ACT: Attentional Control Task; ICT-RST 1 & 2: Inhibitory Control & Task/Rule Switching Task Subtasks 1 (Inhibition) and 2 (Task/Rule switching); ICT-RCT SE: Total number of errors of Inhibitory Control & Task/Rule Switching Task; ICT-RST FS: Failed sets on the Inhibitory Control & Task/Rule Switching Task; CFT: Cognitive Flexibility Task; CFT2 Con. a: Cognitive Flexibility Task part 2—1st level of difficulty; CFT2 Con. b: Cognitive Flexibility Task part 2—2nd level of difficulty; CFT2 Condition c: Cognitive Flexibility Task part 2—3rd level of difficulty; VFT: Visual Fluency Task (Planning, Inhibitory Control, and Visual Fluency); EMT-W Con. a; Episodic Memory Task—Windows (1st level of difficulty); EMT-W Con. b; Episodic Memory Task—Windows (2nd level of difficulty).

**Table 4 diagnostics-13-00338-t004:** (**a**) Direct effects of diagnosis on the R4Alz-R tasks and subtasks’ performance according to Mediation Analysis, between the groups of HC and MCI (* *p* < 0.004). (**b**) Indirect effects of diagnosis—via age and education—on the R4Alz-R tasks’ and subtasks’ performance according to Mediation Analysis, between the groups of SCD and MCI (* *p* < 0.004).

(a)										
									95% Confidence Interval	
Direct Effects					b	SE	z-Value	*p*	Lower	Upper
diagnosis	→	WMCUT S 1			−0.191	0.122	−1.564	0.118	−0.472	0.205
diagnosis	→	WMCUT S 2			−0.604	0.130	−4.646	**<0.001 ***	−0.893	−0.298
diagnosis	→	WMCUT S 3			−1.212	0.337	−3.593	**<0.001 ***	−1.821	−0.699
diagnosis	→	ACT			3.668	1.086	3.377	**<0.001 ***	1.968	5.581
diagnosis	→	ICT-RST 1 & 2			8.102	1.495	5.421	**<0.001 ***	5.175	10.491
diagnosis	→	ICT-RST 1 & 2 SE			0.656	0.154	4.261	**<0.001 ***	0.324	0.937
diagnosis	→	ICT-RST 1 & 2 FS			1.016	0.217	4.689	**<0.001 ***	0.468	1.407
diagnosis	→	CFT			2.562	0.468	5.476	**<0.001 ***	1.703	3.362
diagnosis	→	CFT2 Con. a			0.994	0.280	3.549	**<0.001 ***	0.455	1.503
diagnosis	→	CFT2 Con. b			1.387	0.237	5.853	**<0.001 ***	0.731	1.877
diagnosis	→	CFT2 Con. c			0.988	0.289	3.417	**<0.001 ***	0.160	1.639
diagnosis	→	VFT			−1.302	0.276	−4.723	**<0.001 ***	−1.835	−0.749
diagnosis	→	EMT-W Con. a			1.732	0.374	4.638	**<0.001 ***	1.063	2.488
diagnosis	→	EMT-W Con. b			1.186	0.339	3.499	**<0.001 ***	0.430	1.788
**(b)**										
									**95% Confidence Interval**	
**Indirect Effects**					**b**	**SE**	**z-Value**	** *P* **	**Lower**	**Upper**
diagnosis	→	Age	→	WMCUT S 1	−0.140	0.067	−2.075	0.038	−0.350	−0.043
diagnosis	→	Education	→	WMCUT S 1	−0.040	0.044	−0.898	0.369	−0.203	0.038
diagnosis	→	Age	→	WMCUT S 2	−0.214	0.086	−2.486	0.013	−0.396	−0.076
diagnosis	→	Education	→	WMCUT S 2	−0.030	0.046	−0.653	0.514	−0.200	0.048
diagnosis	→	Age	→	WMCUT S 3	−0.418	0.193	−2.170	0.030	−0.893	−0.092
diagnosis	→	Education	→	WMCUT S 3	−0.120	0.124	−0.964	0.335	−0.507	0.082
diagnosis	→	Age	→	ACT	1.896	0.744	2.546	0.011	0.536	3.870
diagnosis	→	Education	→	ACT	0.341	0.394	0.865	0.387	−0.280	1.628
diagnosis	→	Age	→	ICT−RST 1 & 2	1.917	0.867	2.211	0.027	0.484	4.212
diagnosis	→	Education	→	ICT−RST 1 & 2	0.130	0.514	0.252	0.801	−1.042	1.744
diagnosis	→	Age	→	ICT−RST 1 & 2 SE	0.093	0.071	1.305	0.192	−0.083	0.281
diagnosis	→	Education	→	ICT−RST 1 & 2 SE	0.168	0.083	2.025	0.043	0.017	0.447
diagnosis	→	Age	→	ICT−RST 1 & 2 FS	0.230	0.116	1.980	0.048	0.010	0.560
diagnosis	→	Education	→	ICT−RST 1 & 2 FS	0.124	0.088	1.411	0.158	−0.009	0.415
diagnosis	→	Age	→	CFT	0.624	0.276	2.256	0.024	0.172	1.328
diagnosis	→	Education	→	CFT	0.283	0.193	1.462	0.144	−0.064	0.880
diagnosis	→	Age	→	CFT2 Con. a	0.062	0.120	0.512	0.609	−0.231	0.381
diagnosis	→	Education	→	CFT2 Con. a	0.159	0.114	1.403	0.161	−0.029	0.467
diagnosis	→	Age	→	CFT2 Con. b	0.039	0.101	0.381	0.703	−0.229	0.301
diagnosis	→	Education	→	CFT2 Con. b	0.232	0.120	1.930	0.054	0.025	0.510
diagnosis	→	Age	→	CFT2 Con. c	0.325	0.158	2.052	0.040	0.071	0.800
diagnosis	→	Education	→	CFT2 Con. c	0.044	0.101	0.436	0.663	−0.138	0.334
diagnosis	→	Age	→	VFT	−0.307	0.150	−2.038	0.042	−0.711	−0.056
diagnosis	→	Education	→	VFT	−0.125	0.106	−1.182	0.237	−0.432	0.035
diagnosis	→	Age	→	EMT−W Con. a	0.013	0.159	0.082	0.935	−0.361	0.336
diagnosis	→	Education	→	EMT−W Con. a	0.072	0.131	0.552	0.581	−0.152	0.427
diagnosis	→	Age	→	EMT−W Con. b	−0.034	0.144	−0.235	0.814	−0.427	0.291
diagnosis	→	Education	→	EMT−W Con. b	0.001	0.116	0.010	0.992	−0.337	0.260

Abbreviations: WMCUT S1: Working Memory Capacity and Updating Task—working memory component of short-term store; WMCUT S2: Working Memory Capacity and Updating Task—working memory component of central executive; WMCUT S3: Working Memory Capacity and Updating Task—working memory updating & working memory component of the episodic buffer; ACT: Attentional Control Task; ICT-RST 1&2: Inhibitory Control & Task/Rule Switching Task Subtasks 1 (Inhibition) and 2 (Task/Rule switching); ICT-RCT SE: Total number of errors of Inhibitory Control & Task/Rule Switching Task; ICT-RST FS: Failed sets on the Inhibitory Control & Task/Rule Switching Task; CFT: Cognitive Flexibility Task; CFT Con. a: Cognitive Flexibility Task part 2—1st level of difficulty; CFT Con. b: Cognitive Flexibility Task part 2—2nd level of difficulty; CFT Condition c: Cognitive Flexibility Task part 2—3rd level of difficulty; VFT: Visual Fluency Task (Planning, Inhibitory Control, and Visual Fluency); EMT-W Con. a; Episodic Memory Task—Windows (1st level of difficulty); EMT-W Con. b; Episodic Memory Task—Windows (2nd level of difficulty).

**Table 5 diagnostics-13-00338-t005:** Standard deviations of the R4Alz-R battery’s scores of interest for SCD and HC groups.

R4Alz-R Tasks	Std. Deviation after Min–Max Normalization	AUC
WMCUT S3	0.23166	0.839
ICT-RST 1 & 2	0.26460	0.832
ICT-RST FS	0.31163	0.791
CFT	0.23456	0.876
CFT Con. B	0.31586	0.832

Abbreviations: WMCUT S3: Working Memory Capacity and Updating Task—working memory updating & working memory component of the episodic buffer; ICT-RST 1 & 2: Inhibitory Control & Task/Rule Switching Task—Subtasks 1 (Inhibition) and 2 (Task/Rule switching); ICT-RST FS: Failed sets on the Inhibitory Control & Task/Rule Switching Task; CFT: Cognitive Flexibility Task; CFT Con. B: Cognitive Flexibility Task—2nd level of difficulty.

**Table 6 diagnostics-13-00338-t006:** Diagnostic score classification between SCD and HC.

Total Scores between SCD and HC	Cutoff	AUC	Sensitivity	Specificity	95% CI	*p*-Value
$S_{\Sigma }$	1.0263	0.964	96.6%	95%	0.902–1.000	<0.001
$S_{\Sigma^{2}}$	0.3149	0.972	100%	95%	0.918–1.000	<0.001
$S_{\Sigma^{2}AUC}$	0.2689	0.974	100%	95%	0.923–1.000	<0.001
$S_{\Sigma^{2}AUC^{2}}$	0.2297	0.974	100%	95%	0.923–1.000	<0.001
$S_{\Sigma }^{\sigma }$	3.7292	0.971	100%	95%	0.913–1.000	<0.001
$S_{\Sigma^{2}}^{\sigma }$	**1.2236**	**0.976**	**100%**	**95%**	**0.928–1.000**	**<0.001**
$S_{\Sigma^{2}AUC}^{\sigma }$	1.0479	0.974	100%	95%	0.923–1.000	<0.001
$S_{\Sigma^{2}AUC^{2}}^{\sigma }$	0.8982	0.972	100%	95%	0.918–1.000	<0.001

Abbreviations: SCD: Adults with Subjective Cognitive Decline; HC: Cognitively Healthy Advanced-age Adults (above 50 years); AUC: Area Under the Curve; CI: Confidence interval; Values in bold: the best discrimination formula and its score.

**Table 7 diagnostics-13-00338-t007:** Standard deviations of the R4Alz-R battery’s scores of interest for SCD and MCI groups.

R4Alz-R Tasks	Std. Deviation after Min–Max Normalization	AUC
ICT-RST 1 & 2	0.26460	0.734
ICT-RST SE	0.25238	0.744
CFT Con. A	0.30099	0.692
EMT-W Con. A	0.21215	0.800

Abbreviations: ICT-RST 1 & 2: inhibition and switching 1 & 2; ICT-RST SE: inhibition and switching 1 & 2—switch errors; CFT Con. A: Cognitive Flexibility Task—1st level of difficulty; EMT-W Con. A: Episodic Memory Task—Windows—1st level of difficulty.

**Table 8 diagnostics-13-00338-t008:** Diagnostic score classification between SCD and MCI.

Total Scores between SCD and MCI	Cutoff	AUC	Sensitivity	Specificity	95% CI	*p*-Value
$S_{\Sigma }$	1.5171	0.835	74.2%	82.8%	0.733–0.937	<0.001
$S_{\Sigma^{2}}$	0.8293	0.872	74.2%	86.2%	0.780–0.963	<0.001
$S_{\Sigma^{2}AUC}$	0.5336	0.874	80.6%	82.8%	0.783–0.964	<0.001
$S_{\Sigma^{2}AUC^{2}}$	0.4075	0.873	80.6%	82.8%	0.781–0.964	<0.001
$S_{\Sigma }^{\sigma }$	6.8129	0.860	71%	89.7%	0.768–0.953	<0.001
$S_{\Sigma^{2}}^{\sigma }$	13.9660	0.887	87.1%	79.3%	0.800–0.975	<0.001
$S_{\Sigma^{2}AUC}^{\sigma }$	11.0347	0.889	87.1%	82.8%	0.803–0.976	<0.001
$S_{\Sigma^{2}AUC^{2}}^{\sigma }$	**8.4165**	**0.892**	**90.3%**	**82.8%**	**0.806–0.977**	**<0.001**

Abbreviations: SCD: Adults with Subjective Cognitive Decline; MCI: Adults with Mild Cognitive Impairment; AUC: Area Under the Curve; CI: Confidence interval; Values in bold: the best discrimination formula and its score.

**Table 9 diagnostics-13-00338-t009:** Standard deviations of the R4Alz battery’s scores of interest for HC and MCI groups.

R4Alz-R Tasks	Std. Deviation after Min–Max Normalization	AUC
ICT-RST 1 & 2	0.26460	0.931
CFT Con. A	0.30099	0.798
CFT Con. B	0.31586	0.923
EMT-W Con. A	0.21215	0.857
EMT-W Con. B	0.24559	0.773

Abbreviations: ICT-RCT 1 & 2: inhibition and switching 1 & 2; CFT Con. A: Cognitive Flexibility Task—1st level of difficulty; CFT Con. B: Cognitive Flexibility Task—2nd level of difficulty; EMT-W Con. A: Episodic Memory Task—Windows—1st level of difficulty; EMT-W Con. B: Episodic Memory Task—Windows—2nd level of difficulty.

**Table 10 diagnostics-13-00338-t010:** Diagnostic score classification between HC and MCI.

Total Scores between HC and MCI	Cutoff	AUC	Sensitivity	Specificity	95% CI	*p*-Value
$S_{\Sigma }$	1.7635	0.968	90.3%	95%	0.926–1.000	<0.001
$S_{\Sigma^{2}}$	0.7512	0.973	96.8%	90%	0.935–1.000	<0.001
$S_{\Sigma^{2}AUC}$	0.6167	0.974	96.8%	90%	0.938–1.000	<0.001
$S_{\Sigma^{2}AUC^{2}}$	**0.4804**	**0.977**	**100%**	**90%**	**0.943–1.000**	**<0.001**
$S_{\Sigma }^{\sigma }$	6.5582	0.971	93.5%	90%	0.934–1.000	<0.001
$S_{\Sigma^{2}}^{\sigma }$	3.0873	0.968	96.8%	90%	0.926–1.000	<0.001
$S_{\Sigma^{2}AUC}^{\sigma }$	2.5263	0.973	96.8%	90%	0.935–1.000	<0.001
$S_{\Sigma^{2}AUC^{2}}^{\sigma }$	2.0760	0.976	96.8%	90%	0.942–1.000	<0.001

Abbreviations: HC: Cognitively Healthy Advanced-age Adults (above 50 years); MCI: Adults with Mild Cognitive Impairment; AUC: Area Under the Curve; CI: Confidence interval; Values in bold: the best discrimination formula and its score.

## Data Availability

Data are not available due to privacy reasons and ethical restrictions.

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
