# Peer review of "R4Alz-Revised: A Tool Able to Strongly Discriminate ‘Subjective Cognitive Decline’ from Healthy Cognition and ‘Minor Neurocognitive Disorder’"

_diagnostics, 2023, doi:10.3390/diagnostics13030338_

Round 1
Reviewer 1 Report
This is a timely and interesting study focusing on better screening and identification of SCD in healthy adults of advanced age. Despite the relevance of this for scientists and clinicians working in the area of cognitive sciencs, I have some concerns.
1. First, regarding the paper's focus, I have confused somehow. As you mentioned in your earlier manuscript (Ref 9), the R4ALZ battery is capable of neuropsychological assessment of people with SCD, and also it helped discriminate SCD from HOA, then what is your rationale for extending this battery apart from age and education variables?
2. Some essential pieces of literature are missing in this paper; you may cross-check them once. (Ibnidris, A., Robinson, J.N., Stubbs, M. et al. Evaluating measurement properties of subjective cognitive decline self-reported outcome measures: a systematic review. Syst Rev 11, 144 (2022).) (Clara Li, et al; 2022)
3. Authors may explain this tool's cross-cultural validity and use as we consider populations from developing countries like India, and you can incorporate this issue in limitation.
4. In terms of the introductory paragraph, the references are quite old (Ref 1-8). Is there newer data on this, especially in the progression of SCD into MCI, and tools to differentiate SCD from HOA, could provide more strength for this paper?
Author Response
Dear reviewer,
Below you may find our responses regarding your comments, after the first review. The format below adheres to the format:
(R)eviewer(Reviewer number)(C)omment(Comment number)
(R)eviewer(Reviewer number)/(R)esponse
#1 Reviewer’s Comments and Suggestions for Authors
This is a timely and interesting study focusing on better screening and identification of SCD in healthy adults of advanced age. Despite the relevance of this for scientists and clinicians working in the area of cognitive sciencs, I have some concerns.
R1C1: First, regarding the paper's focus, I have confused somehow. As you mentioned in your earlier manuscript (Ref 9), the R4ALZ battery is capable of neuropsychological assessment of people with SCD, and also it helped discriminate SCD from HOA, then what is your rationale for extending this battery apart from age and education variables?
R1R1: Dear reviewer, thank you for the comment. As described to the introduction section “Besides the fact that the R4Alz battery seems to have an excellent discrimination ability between healthy cognition and SCD, it has only a fare discrimination ability among SCD and MCI”. Therefore, we considered that it was very crucial to raise the discriminant validity between SCD and MCI from fare to excellent. For this reason, we added extra tasks, not only executive control which had already been proven to have excellent discrimination from the last study but also episodic memory tasks, since as described in the discussion section, the course of neurodegeneration might begin with cognitive changes in executive function abilities, when the deficits are measurable and objective as occurs in amnestic MCI, the memory problems seem to be more prominent.
R1C2: Some essential pieces of literature are missing in this paper; you may cross-check them once. (Ibnidris, A., Robinson, J.N., Stubbs, M. et al. Evaluating measurement properties of subjective cognitive decline self-reported outcome measures: a systematic review. Syst Rev 11, 144 (2022).) (Clara Li, et al; 2022)
R1R2: We have added to the discussion section a sentence (page 24, lines 853-854) regarding the validity of the self-questionnaires in SCD, referencing the paper of Ibnidris at al, 2022. Unfortunately, the Clara Li’s article was not possible to be found, since the reviewer does not provide more information such as the title of the paper. We would appreciate it if the reviewer could give us more details on this publication.
R1C3: Authors may explain this tool's cross-cultural validity and use as we consider populations from developing countries like India, and you can incorporate this issue in limitation.
R1R3: We thank the reviewer for this really essential comment. Based on the fact that R4Alz-R is a battery that mainly utilizes visual-spatial abilities such as colors, windows, or photos of animals,, we consider that it is also free of cultural effects, as no cultural cues are whatsoever used in the inherent tests. We have added a related paragraph in the conclusion section on page 27, lines 1023-1026.
R1C4: In terms of the introductory paragraph, the references are quite old (Ref 1-8). Is there newer data on this, especially in the progression of SCD into MCI, and tools to differentiate SCD from HOA, that could provide more strength for this paper?
R1R4: With all the respect to the reviewer, the reference [1] refers to Reisberg and colleagues who were the first that introduced the concept of SCD, therefore we believe that even though this is an old reference, it should be mentioned. Similarly, the reference [3] refers to Jessen and colleagues in 2014, being the first SCD working group that proposed specific criteria for SCD, therefore we believe that it should also be mentioned at the beginning of the introduction section. References 2 and 4 which were quite old were changed to newer references, referring to the progression from SCD to MCI. References no 5-8 refer to evidence regarding SCD and MCI from 2019 to 2021, which are recent and therefore we decided to keep them.
Reviewer 2 Report
In this study, Poptsi et al. extend the R4Alz battery by designing tasks for enhancing the ability to discriminate SCD from healthy cognition and MCI.
This is a well-conducted study with a good design. The sample size is relatively small. However, the methods are clearly spelled out and the results are presented effectively. The discussion is reasonably critical and concise and addresses adequately the limitations of the study. The topic is of great importance and emphasizes the importance of designing tools to discriminate SCD from MCI and controls.
I have only some minor comments:
-In the abstract, please explain what R4A1z is.
-Line 783: “that supports the idea that” should not be in italics.
-Comments on the total score creation process should be moved to the discussion
-The limitation section should be moved before the conclusion
-In the discussion, the authors may comment on the fact that biomarkers of disease were not available, especially for amyloid and tau. Further studies could explore the discrimination abilities of this interesting tool in the biological Alzheimer spectrum (SCD and MCI with biological AD, and SCD and MCI with non-AD). They can also comment on the possibility of using this tool in population-based studies on SCD.
Author Response
Dear reviewer,
Below you may find our responses regarding your comments, after the first review. The format below adheres to the format:
(R)eviewer(Reviewer number)(C)omment(Comment number)
(R)eviewer(Reviewer number)/(R)esponse
#2 Reviewer’s Comments and Suggestions for Authors
In this study, Poptsi et al. extend the R4Alz battery by designing tasks for enhancing the ability to discriminate SCD from healthy cognition and MCI.
This is a well-conducted study with a good design. The sample size is relatively small. However, the methods are clearly spelled out and the results are presented effectively. The discussion is reasonably critical and concise and addresses adequately the limitations of the study. The topic is of great importance and emphasizes the importance of designing tools to discriminate SCD from MCI and controls.
I have only some minor comments:
R2C1: In the abstract, please explain what R4A1z is.
R2R1: We have added at the abstract section a small sentence which briefly explains what the original R4Alz battery is (page 1, lines 20-23).
R2C2: Line 783: “that supports the idea that” should not be in italics.
R2R2: We removed the italics from the sentence.
R2C3: Comments on the total score creation process should be moved to the discussion
R2R3: We thank the reviewer for the suggestion. We have indeed moved “the comments on the total score creation” to the discussion section.
R2C4: The limitation section should be moved before the conclusion
R2R4: The limitation section was moved as suggested.
R2C5: In the discussion, the authors may comment on the fact that biomarkers of disease were not available, especially for amyloid and tau. Further studies could explore the discrimination abilities of this interesting tool in the biological Alzheimer spectrum (SCD and MCI with biological AD, and SCD and MCI with non-AD). They can also comment on the possibility of using this tool in population-based studies on SCD.
R2R5: We would like to thank the reviewer for the suggestions. We have integrated the aforementioned comments to the limitation section as well as to the conclusion section (page 26, lines 997-1001 and page 27, lines 1026-1028).